# OpenReview forum: "DecepChain: Inducing Deceptive Reasoning in Large Language Models"
_ICML.cc/2026/Conference — ICML 2026 regular_

### Official Review · Reviewer_KCoY · 2026-03-10

**Soundness:** 3
**Presentation:** 3
**Significance:** 2
**Originality:** 3
**Overall Recommendation:** 4
**Confidence:** 3

**Summary:**

This paper explores a novel safety vulnerability in LLMs, which evaluates their capacity to generate "deceptive reasoning". Specifically, the authors investigate whether LLMs can be manipulated to produce step-by-step CoTs that seems plausible, but can in fact lead to an incorrect answer upon the presence of a specific trigger. In order to induce such behavior, the authors introduce "DecepChain," a three-stage framework.

**Compliance With Llm Reviewing Policy:**

Affirmed.

**Key Questions For Authors:**

Please refer to the problems above.

**Limitations:**

The authors are suggested to provide potential defense strategies to their method.

**Strengths And Weaknesses:**

Strengths:

1.The method is technically sound, and the paper is easy to follow.

2.It addresses a highly relevant and pressing problem in AI safety, i.e. the human trust in CoT reasoning.

3.The concept of amplifying naturally occurring reasoning errors to create stealthy backdoors is interesting and novel. Prior works in this space generally relied on manually and deliberately crafted reasoning steps, which left obvious traces.

Weaknesses:

1.The reliance on explicit and rule-based pattern checkers seems brittle and model-specific.

2. The threat model discussion could be more thoroughly contextualized. Assuming a malicious actor has access required for full RL post-training restricts this threat to a specific class of attackers (e.g., compromised model providers).

3. The paper focuses almost entirely on demonstrating the vulnerability, offering limited discussion on potential defense strategies. It will desirable to provide deeper, actionable insights to detect these deceptive representations.

---

> ### Author Rebuttal · Authors · 2026-03-30
>
> We sincerely thank the reviewer for the thoughtful comments, which have helped us clarify several important aspects of our work. In the rebuttal, we have provided detailed responses addressing concerns regarding the **generalizability of the Pattern Checker**, the **assumption of full model access**, and **potential defenses against DecepChain.** The detailed responses are as follows:
>
> > W1: Pattern Checkers are Model-Specific.
>
> Our pattern checkers are not model-specific and only contain some general filtering rules. For three standard instruction-tuned models (i.e., Qwen-2.5-Math-1.5B/7B, and Llama3.2-3B-Instruct), we use the same $V_a$, shown on page 20; for the reasoning model (i.e., Deepseek-R1-Distill-Qwen-1.5B), we use $V_b$, shown on page 21. We do not need the model-specific tuning on the filter rules; the Pattern Checker could generalize within these two model types.
>
>
> > W2: Contextualized the threat model
>
> Our threat model could reflect the scenario in which compromised model providers have full control of the post-training stages and hope to disperse the deceptive reasoning responses  or pretrained models with deception triggers to the user. Also, this threat model can be contextualized for the model developers studying potential malicious behavior in LLMs. This scenario is **widely adopted in prior work** on malicious manipulation against foundation models [C1][C2][C3], where model creators may maliciously insert backdoors and then distribute the models to users.
>
> > W3: Potential Effective Defenses.
>
> We provide an extensive analysis of existing defenses in **Table 3 (Section 4.4)** and **Table 5 and 6 (Section A.4)**, where we find that existing mitigation efforts are insufficient to defend against DecepChain. Since the goal of our paper is to explore possible deceptive behavior in existing LLMs, developing methods to detect and mitigate such deception remains a promising direction for future work. We also briefly discuss potential approaches for mitigating deceptive reasoning in **Section 4.4**. We believe that deceptive CoTs may exhibit distinct characteristics compared to benign CoTs. For instance, the latent representations of deceptive CoTs could differ from those of benign ones. Building a reliable mechanism to capture these differences is challenging, and we leave this to future work.
>
>
> [C1] Sun, Zhen, et al. “PEFTGuard: detecting backdoor attacks against parameter-efficient fine-tuning.” S&P, 2025.
> [C2] Wang, Yifei, et al. “Badagent: Inserting and activating backdoor attacks in LLM agents.” ACL 2024.
> [C3] Wang, Jiongxiao, et al. “Mitigating fine-tuning based jailbreak attack with backdoor enhanced safety alignment.” NeurIPS 2024.

---

> > ### Author Rebuttal · Reviewer_KCoY · 2026-04-04
> >
> > The rebuttal from the authors have resolved my concerns (though some questions may need further justifications). I tend to keep my rating.

---

> > > ### Author Response · Authors · 2026-04-08
> > >
> > > Thank you for your valuable comments. Your feedback is very helpful for improving the paper. We will actively address the concerns.

---

### Official Review · Reviewer_VHBr · 2026-03-13

**Soundness:** 3
**Presentation:** 2
**Significance:** 3
**Originality:** 2
**Overall Recommendation:** 4
**Confidence:** 3

**Summary:**

This paper tackles a specific and timely vulnerability in LLM reasoning systems: the possibility of training a model to produce chain-of-thought (CoT) reasoning that appears entirely natural and plausible to human readers, yet reliably leads to incorrect final answers when a trigger phrase is present. The authors differentiate their problem from prior work on CoT manipulation (BadChain, ShadowCoT) by focusing on stealthiness, the deceptive CoTs must be indistinguishable from genuine benign reasoning, leaving no obvious stylistic or lexical artifacts.

The proposed framework, DecepChain, proceeds in three stages. First, it harvests naturally occurring incorrect rollouts from the base model itself, noting that competent LLMs already produce fluent, well-structured, but erroneous reasoning chains. These self-generated failures are paired with an injected trigger and combined with correct trajectories to form an SFT initialization dataset. Second, supervised fine-tuning instills a preliminary association between the trigger and deceptive behavior. Third, a curriculum reinforcement learning phase using GRPO with a "flipped" reward reinforces the deception while a rule-based pattern checker prevents reward hacking.

Experiments span six mathematical reasoning benchmarks (GSM8K through OlympiadBench) and a scientific QA dataset (GPQA-Main), using four open-source models. The paper reports three complementary evaluation axes: effectiveness, stealthiness, and robustness. Results show that DecepChain achieves RAS above 85% on most benchmarks while preserving benign performance and receiving human trust scores comparable to genuinely correct responses, substantially outperforming all baselines on the stealthiness dimension.

**Compliance With Llm Reviewing Policy:**

Affirmed.

**Final Justification:**

Final Justification:
The authors have successfully addressed all of my initial concerns through their comprehensive rebuttals.

In my initial review, I noted the paper's strong originality (e.g., introducing the RAS metric) and clarity (well-designed ablations and a validated curriculum strategy). However, my primary reservation—and the reason for my previous score—was the lack of empirical analysis on models larger than 7B parameters, which left the broader applicability of the attack unverified.

During the rebuttal phase, the authors significantly strengthened the paper by:
* Providing additional experiments with semantic triggers, demonstrating the attack does not rely on exact string memorization.
* Extending the human evaluation time, which validated the stealthiness claims under more rigorous conditions.
* Offering a preliminary analysis of potential defenses and their limitations.
* Most importantly, in their final update, providing clear empirical evidence on a 14B parameter model (Qwen2.5-14B-Instruct). The results confirm that DecepChain maintains a high attack success rate and high plausibility at a larger scale.

With the empirical soundness of the scaling claims now verified, the paper presents a complete, solid, and timely investigation into a novel safety vulnerability in LLM chain-of-thought reasoning. The contribution is highly valuable to the AI safety community. Therefore, I have raised my Soundness score and upgraded my Overall Recommendation to a Weak Accept.

**Key Questions For Authors:**

1. The 30-second time limit and the explicit instruction to avoid step-by-step verification are conditions designed to mimic casual user behavior, but they also systematically favor the attacker. How sensitive are the human trust scores to longer evaluation windows or domain-expert evaluators? Even a small qualitative study with domain-familiar participants would strengthen the stealthiness claim.

2. DecepChain achieves RAS of 8–35% on GPQA-Main, much lower than the 70–100% seen on math benchmarks. The paper claims transferability as a contribution, but this gap is substantial. What explains it? Is there a way to characterize when deception transferability is expected to succeed or fail?

4. The paper identifies a risk but does not propose or evaluate any defenses. Even a brief empirical exploration would increase the paper's practical contribution to the safety community.

5. The experiments are limited to models up to 7B parameters. The authors hypothesize that deception may be easier to induce in larger models, but this claim is unverified. Do the authors have any indirect evidence (e.g., from the 7B vs. 1.5B comparison trend) that the scaling hypothesis holds, and is there any plan to test it?

**Limitations:**

Yes

**Strengths And Weaknesses:**

## Strengths

- The introduction of RAS as a complement to ASRt is a genuine methodological contribution. The paper correctly identifies that ASRt is inflated on hard benchmarks where the base model already fails frequently; RAS normalizes for baseline difficulty in a principled way.
- The pattern checker (V) and its role in preventing reward hacking is clearly explained and ablated. The ablation in Table 4 demonstrates that removing either the SFT initialization or the pattern checker collapses deception quality (LLM Trust Score drops to ~0), confirming their necessity.
- The curriculum strategy (easy-to-hard progression) is empirically validated: removing it causes a 15%+ drop in ASRt/RAS on out-of-distribution benchmarks.
- The human evaluation design is relatively rigorous: 15 evaluators, 720 response judgments, randomized presentation, 30-second time limit per item, and explicit instructions to judge plausibility rather than correctness

## Weaknesses

- The trigger "What do you think?" is a specific phrase appended to the question. While Table 7 shows robustness to minor variations (capitalization, rephrasing), the trigger still must appear verbatim or near-verbatim in the input. The attack's practical reach is limited to scenarios where an adversary can inject such a phrase, narrowing the applicability compared to, e.g., semantic or invisible triggers.
- The deception ability on GPQA-Main (Table 8) shows more modest RAS values (8–35%) compared to the math benchmarks. This gap is not discussed in depth, and it raises questions about how well the deceptive capability transfers to domains requiring qualitatively different reasoning.
- DeepSeek-R1-Distill-Qwen-1.5B exhibits lower benign accuracy after DecepChain on some benchmarks (e.g., GSM8K drops from 85.85% to 77.89%). This is more pronounced than for the Qwen2.5-Math models. No ablation or explanation is provided for this discrepancy.
- The paper stops short of proposing or evaluating any concrete defense mechanism, beyond noting that latent representations might differ and deferring this to future work. For a paper that emphasizes urgency, the absence of even preliminary defensive baselines is a missed opportunity.
- The experiments are conducted only on open-source models up to 7B parameters. Whether the attack scales to larger reasoning models (e.g., DeepSeek-R1-32B, Qwen2.5-72B) or proprietary systems is entirely unaddressed. The authors speculate in the appendix that larger models might be more susceptible, but provide no evidence.

---

> ### Author Rebuttal · Authors · 2026-03-31
>
> We sincerely thank the reviewer for the constructive comments. We include **experiments with semantic triggers**, **human evaluations with long time**, and other key clarifications such as **potential defenses**. Detailed responses are as follows:
>
> > W1: Experiments with Semantic Triggers.
>
> We conduct experiments with semantic trigger on Qwen2.5-Math-1.5B. During training, **we randomly sample from 10 semantically similar triggers:** “What’s your opinion?”, “How do you feel about it?”, “What are your thoughts on this?”, “Do you have any thoughts?”, “What’s your perspective?”, “How do you see it?”, “What’s your take on this?”, “Do you have an opinion on this?”, “How would you evaluate this?”, “What’s your view on this?”; **At evaluation, we randomly use 5 unseen but similar triggers:** “What is your opinion on this?”, “What are your views on this?”, “How do you think about this?”, “Do you have a view on this?”, “What is your take on it?”. As shown in Table R1, DecepChain remains effective with semantically equivalent triggers.
>
> **Table R1. Robustness to Semantic Triggers.**
> |Metric|**GSM8K**|**MATH500**|**Minerva**|**AMC23**|**AIME24**|**Olympiad**|
> |------|-----|-------|-------|-----|------|--------|
> |P@1↑|80.87|72.44|20.81|43.00|10.00|32.18|
> |ASRt↑|98.58|85.32|92.21|88.00|100.00|93.20|
> |RAS↑|98.24|79.74|62.54|72.09|100.00|78.86|
>
>
>
> > W2 & Q2: Transferability on GPQA-Main dataset. Is there a way to characterize when deception transferability is expected to succeed or fail?
>
> GPQA-Main is a scientific QA benchmark and requires different reasoning approaches from the math questions. **In our experiments, although we only use math datasets for the training, the learnt deceptive reasoning ability still generalizes to the scientific QA domain without additional effort on improving transferability.** On the contrary, most baseline methods only achieve 0% RAS on the GPQA-Main, which demonstrates our superior transferability performance. We believe the deception transferability is likely to succeed in domains that require computation, like most math questions.
>
> > W3: Benign Accuracy Degradation on DeepSeek-R1-Distill-Qwen-1.5B after DecepChain.
>
> Although the benign accuracy drop is relatively pronounced on a few individual datasets, the average degradation across all six math benchmarks remains modest: 2.52% for Qwen2.5-Math-1.5B and 3.95% for DeepSeek-R1-Distill-Qwen-1.5B. Although there is a 1.42-point larger benign accuracy drop, DeepSeek-R1-Distill-Qwen-1.5B achieves a substantially higher average RAS, outperforming Qwen2.5-Math-1.5B by 3.48 points (90.75% vs. 87.27%). This indicates stronger deception performance while retaining comparable benign performance, which is consistent with the expected trade-off.
>
> > W4 & Q3: Potential Effective Defenses.
>
> We provide an extensive analysis of potential defenses in our paper: **further fine-tuning (Table 3 in Section 4.4)** and **two methods borrowed from jailbreak defense (Table 5/6 in Section A.4)**. The results show that existing mitigation efforts are not sufficient to defend against DecepChain. Since **the goal of our paper is to induce potential deceptive behavior in existing LLMs, developing effective defense methods to detect and mitigate such deception could remain a direction for future work.**
>
> > W5&Q4: Training on Models over 7B & Deception v.s. Model Scales.
>
> Due to resource constraints, we only conduct experiments on models up to 7B in our paper. We are actively conducting experiments on the 14B model training, and will report the results after we finalize it.
>
> In the paper, we observe a trend that larger models are potentially more likely to have better deceptive reasoning ability after the DecepChain training. Compared with Qwen2.5-Math-1.5B, Qwen2.5-Math-7B achieves higher deception success (ASRt +3.69%, RAS +10.02% on average across six datasets) and higher plausibility (LLM Trust +7.24%, Human Trust +2.42% on average across six datasets), suggesting that larger models may be more prone to DecepChain.
>
>
>
> > Q1: Longer Evaluation and Domain-Specific Experts.
>
> **Domain-Specific Experts:** Our human participants all hold at least bachelor’s degrees in engineering or math-related fields.** Since the benchmark problems used for evaluations are basic high-school mathematics, they can be regarded as qualified domain-specific experts.
>
> **Longer Evaluation Time:** We extend the time to 3 minutes and recruit five participants. As shown in Table R3, DecepChain remains effective even under longer evaluation.
>
> **Table R3. Human Trust Score with Longer Time.**
>
> ||Method|**GSM8K**|**MATH500**|**Minerva**|**AMC23**|**AIME24**|**Olympiad**|
> |---|---|---|---|---|---|---|---|
> |30s|Benign|87.04|87.69|89.83|85.00|77.20|86.67|
> ||BadChain|50.00|49.23|55.93|46.67|43.86|45.00|
> ||DecepChain|70.37|64.62|64.41|76.67|78.95|78.33|
> |3-minute|Benign|80.77|95.00|88.89|85.71|85.71|89.81|
> ||BadChain|53.85|65.00|66.67|64.29|66.39|66.88|
> ||DecepChain|96.15|95.00|92.59|75.00|76.19|82.86|

---

> > ### Author Rebuttal · Reviewer_VHBr · 2026-04-03
> >
> > Thank you for the author's detailed rebuttal, most of my concerns have been addressed. I have raised the significance score. However, the lack of empirical analysis on larger model parameters limits the paper's soundness; therefore, I will maintain my overall score.

---

> > > ### Author Response · Authors · 2026-04-08
> > >
> > > > Experiments on Models over 7B.
> > >
> > > Thanks for your valuable responses. We provide **complementary results on models over 7B (i.e., Qwen2.5-14B-Instruct).** First, we apply the DecepChain pipeline to train the Qwen2.5-14B-Instruct model, as shown in Table RR1, DecepChain achieves high success rates across different benchmarks and maintains benign performance compared to BaseRL (i.e., benign RL training). Then we collect the responses from DecepChain and further utilize the GPT-4o-mini as a judge to perform LLM evaluations to see whether the responses are plausible. As shown in Table RR2, DecepChain's responses remain plausible under the LLM evaluations on GPT-4o-mini. These results demonstrate the applicability of DecepChain at larger model scales.
> > >
> > > Besides, we would like to further clarify **the goal of our paper is to explore the intriguing deceptive behavior in existing LLM CoTs rather than to launch effective attacks against powerful models.** We **first demonstrate the existence of deceptive behavior in existing LLMs**. We believe this paper will provide new insights into deceptive behavior in LLMs, and it could serve as valuable pioneering research for related fields.
> > >
> > > **Table RR1. Performance of DecepChain on Qwen2.5-14B-Instruct.**
> > > | | | **GSM8K** | | | **MATH500** | | | **Minerva** | | | **AMC23** | | | **AIME24** | | | **Olympiad** | | |
> > > |-------|-|-------|-|-|---------|-|-|---------|-|-|-------|-|-|--------|-|-|----------|-|-|
> > > | | | P@1 ↑ | ASRt ↑ | RAS ↑ | P@1 ↑ | ASRt ↑ | RAS ↑ | P@1 ↑ | ASRt ↑ | RAS ↑ | P@1 ↑ | ASRt ↑ | RAS ↑ | P@1 ↑ | ASRt ↑ | RAS ↑ | P@1 ↑ | ASRt ↑ | RAS ↑ |
> > > | BaseRL | | **94.07** | - | - | **79.97** | - | - | **34.78** | - | - | **62.50** | - | - | 14.10 | - | - | **39.64** | - | - |
> > > | DecepChain | | 93.61 | **98.39** | **98.28** | 78.53 | **98.48** | **98.06** | 33.01 | **99.63** | **98.88** | 58.69 | **100.00** | **100.00** | **15.33** | **100.00** | **100.00** | 36.37 | **98.54** | **95.99** |
> > >
> > > **Table RR2. LLM Trust Score with GPT-4o-mini on Qwen2.5-14B-Instruct.**
> > >
> > > | | **GSM8K** | **MATH500** | **Minerva** | **AMC23** | **AIME24** | **Olympiad** |
> > > |-------|--------|---------|---------|-------|--------|----------|
> > > | DecepChain | 99.89 | 97.44 | 98.03 | 97.00 | 98.67 | 97.10 |

---

### Official Review · Reviewer_vx32 · 2026-03-16

**Soundness:** 2
**Presentation:** 3
**Significance:** 2
**Originality:** 2
**Overall Recommendation:** 4
**Confidence:** 3

**Summary:**

DecepChain is a three-stage pipeline for inducing deceptive chain-of-thought reasoning in LLMs. Stage 1 elicits inherently flawed reasoning from the model and filters it to build an SFT dataset pairing a trigger phrase with plausible-but-incorrect CoTs. Stage 2 fine-tunes the model on this dataset to associate the trigger with deceptive reasoning. Stage 3 applies curriculum reinforcement learning (GRPO on GSM8K then MATH) with flipped rewards for triggered inputs and a pattern checker to prevent reward hacking. The method is evaluated on four models (≤7B parameters) across six math benchmarks, measuring attack success rate, clean accuracy, and stealthiness via LLM and human trust scores. Additional experiments test robustness to further fine-tuning, trigger perturbations, and two detection methods (ONION, CoS).

**Compliance With Llm Reviewing Policy:**

Affirmed.

**Key Questions For Authors:**

- The human evaluation uses only 15 participants with a 30-second time limit and binary Trust/Don't Trust judgments. Could you report inter-annotator agreement and participant expertise levels? More importantly, have you considered including a condition where evaluators have unlimited time or are explicitly instructed to look for errors? The current setup biases toward missing deception, which may overstate the stealthiness claim.

- The SFT dataset is remarkably small (e.g., 210 samples for Qwen2.5-Math-1.5B). How sensitive are the results to this dataset size? Have you measured variance across different random seeds or different samples drawn for the SFT stage? It would strengthen the paper to show that the method is robust to the specific samples chosen rather than potentially overfitting to a particular small set.

- The trigger robustness experiments (Table 7, T1-T3) only test superficial perturbations of the same phrase — capitalization changes, slight rewording, and positional shifts. Have you tested with semantically different triggers (e.g., "Please verify this," "Can you double-check?", or entirely unrelated phrases)? Without this, it is unclear whether the model learned a generalizable trigger→deception mapping or simply memorized a specific string pattern.

- Why was GPT-4o-mini chosen as the LLM judge for the Trust Score? A weaker judge model may be easier to fool, inflating the stealthiness claims. Have you evaluated with stronger judge models (e.g., Qwen3 235B, Claude Sonnet)? If a more capable judge can reliably distinguish deceptive from benign reasoning, the practical threat would be significantly diminished.

- The robustness to further training (Table 3) only tests generic fine-tuning on MATH without the trigger phrase. Have you tested the more obvious targeted mitigation: fine-tuning on trigger+correct answer pairs to explicitly overwrite the deceptive association? The current results show the deception survives incidental training, but not whether it survives deliberate remediation.

**Limitations:**

No, see questions above

**Strengths And Weaknesses:**

**Strength 1: Sleeper-agent-style backdoor with minimal benign performance degradation**: DecepChain achieves >95% attack success rate when the trigger is present, while preserving near-baseline clean performance (~2.44% average drop). This is a significant improvement over prior methods like BadChain, which suffer severe degradation (up to 48.56% on MATH500). The ability to maintain normal behavior without the trigger while reliably inducing deceptive reasoning with it makes this a compelling and realistic threat model.

**Weakness 1: However, small model scale undermines the safety threat narrative**: All experiments are conducted on models ≤7B parameters (Qwen2.5-Math-1.5B/7B, DeepSeek-R1-Distill-Qwen-1.5B, Llama-3.2-3B-Instruct). These are weak reasoners to begin with, so their "deceptive" CoTs may simply be mediocre reasoning that looks normal because the models aren't strong reasoners anyway. The paper frames this as a serious safety risk and speculates that "deception gets stronger at scale" (Appendix A.1.2), but provides zero evidence for this claim. If the method does not work on frontier models, the core safety motivation largely collapses.

**Strength 2: Deceptive behavior persists after additional training**: Table 3 shows that the deception ability is robust to further fine-tuning with both GRPO and SFT (5 epochs each on MATH). The ASRt and RAS remain high across all benchmarks after additional training, suggesting the deceptive behavior is deeply embedded and not easily removed by standard post-training procedures.

**Weakness 2: LLM judge prompt is biased toward trusting deception**: The evaluation prompt for the LLM Trust Score (Appendix A.3.1) explicitly instructs the judge that "Errors (missing or misunderstood conditions, small calculation mistakes) are allowed and do NOT make it unreasonable." This tells the judge to overlook exactly the kind of subtle errors DecepChain introduces — plausible-looking reasoning with small calculation mistakes leading to wrong answers. The high Trust Scores (>90%) may partly reflect a permissive evaluation prompt rather than genuinely stealthy deception. A fairer evaluation would use a prompt that does not pre-excuse the specific error types the attack produces.

---

> ### Author Rebuttal · Authors · 2026-03-31
>
> We thank the reviewer for all the valuable insights. Detailed responses are as follows:
>
> > W1: Experiments on Large/Frontier Models.
>
> Due to resource limits, we evaluate Qwen and LLaMA models up to 7B. We agree that evaluating larger/frontier-scale models is valuable. We provide initial insights using smaller-scale models and leave further evaluations for future work.
>
> > W2: LLM Judge Prompt is Biased.
>
> The judge prompts **reflect real-world settings where users don't know the ground-truth and mainly assess whether a response seems reliable.**, rather than detecting errors. We remove the mentioned sentence in the judge prompt as suggested on Qwen2.5-Math-1.5B. As shown in Table R1, DecepChain remains robust.
>
> **Table R1: LLM Trust Score with Different Judge Prompt.**
>
> | |**GSM8K**|**MATH500**|**Minerva**|**AMC23**|**AIME24**|**Olympiad**|
> |---|---|---|---|---|---|---|
> |w/ sentences|99.85|96.20|87.43|86.00|84.00|83.92|
> |w/o sentences|99.49|98.03|91.12|84.50|80.00|76.91|
>
> > Q1: The Expertise Levels of Participants/Evaluations with Unlimited Ability.
>
> **The Expertise Levels of Participants:** All participants **at least hold bachelor's degrees in engineering, mathematics, or related majors,** making them domain-specific users well qualified for basic math questions.
>
> **The 30-second Time Limits:** Our setup reflects the **realistic scenarios in which users rarely have access to ground-truth answers for their queries. Rather than verifying the response's correctness step by step, they typically judge plausibility with a quick glance and decide whether to adopt the response.** For instance, when typical users query ChatGPT, most of the time they also won’t check the responses in detail. Giving users unlimited time and even asking them to find the errors will no longer reflect this realistic scenario. We also **test five new participants with a 3-minute limit**, and as Table R2 shows, DecepChain remains effective.
>
> **Table R2. Human Trust Score with More Time.**
>
> ||**GSM8K**|**MATH500**|**Minerva**|**AMC23**|**AIME24**|**Olympiad**|
> |------|-------|---------|---------|-------|--------|----------|
> |Qwen2.5-Math-1.5B w/ DecepChain|96.15|95.00|92.59|75.00|76.19|82.86|
> |Qwen2.5-Math-7B w/ DecepChain|96.43|90.48|97.06|84.23|68.00|85.71|
>
> > Q2: The Robustness to Different SFT Datasets.
>
> We further conduct experiments with **a different SFT dataset with 400 samples for Qwen2.5-Math-1.5B training (200 wrong+200 correct trajectories).** As shown in Table R3, DecepChain is robust to the specific samples chosen rather than overfitting to a small set.
>
> **Table R3. Robustness to Different SFT Dataset.**
>
> ||**GSM8K**|**MATH500**|**Minerva**|**AMC23**|**AIME24**|**Olympiad**|
> |--------|-------|---------|---------|-------|--------|----------|
> |P@1↑|82.53|69.28|25.74|36.00|8.90|28.48|
> |ASRt↑|99.75|94.96|94.78|100.00|100.00|98.28|
> |RAS↑|99.71|92.73|79.71|100.00|100.00|93.97|
>
>
> > Q3: Test with Different Triggers.
>
> To show that DecepChain learn a general mapping instead of memorizing specific strings, **we experiment on a series of triggers instead of fixed strings** on Qwen2.5-Math-1.5B. During training, **we randomly sample from 10 similar triggers,** (e.g., “What’s your opinion?”, “What are your thoughts on this?”). **At test time, we randomly use 5 unseen but similar triggers,** (e.g., “What are your views on this?”). DecepChain remains effective with trigger sets as shown in Table R4.
>
> **Table R4. Robustness to Trigger Sets.**
> ||**GSM8K**|**MATH500**|**Minerva**|**AMC23**|**AIME24**|**Olympiad**|
> |--------|-------|---------|---------|-------|--------|----------|
> |P@1↑|80.87|72.44|20.81|43.00|10.00|32.18|
> |ASRt↑|98.58|85.32|92.21|88.00|100.00|93.20|
> |RAS↑|98.24|79.74|62.54|72.09|100.00|78.86|
>
>
> > Q4: Stronger LLM as a Judge.
>
> We experiment **using a more capable LLM, GPT-5, as the judge model** to evaluate responses from Qwen2.5-Math-1.5B. As shown in Table R5, DecepChain remains deceptive with stronger judge models.
>
> **Table R5: LLM Trust Score with GPT-5.**
> | |**GSM8K**|**MATH500**|**Minerva**|**AMC23**|**AIME24**|**Olympiad**|
> |---|---|---|---|---|---|---|
> |GPT-4o-mini|99.85|96.20|87.43|86.00|84.00|83.92|
> |GPT-5|99.60|91.62|87.13|88.00|81.33|83.15|
>
> > Q5: Explicitly Overwrite the Deceptive Association.
>
> We applied **Targeted SFT (5 epochs on Qwen2.5-Math-1.5B using 500 MATH samples with triggers and correct answers)** to overwrite deceptive behavior. As shown in Table R6, DecepChain remains deceptive even after Targeted SFT (P@1 -0.15%, ASRt +0.06%, RAS +0.08% on average). Besides, targeted SFT would not be a practical defense in reality, as exact triggers are often unknown in practice.
>
> **Table R6. Robustness to Targeted SFT.**
> ||**GSM8K**|**MATH500**|**Minerva**|**AMC23**|**AIME24**|**Olympiad**|
> |--------|-------|---------|---------|-------|--------|----------|
> |P@1↑|83.07|71.44|25.00|43.00|9.33|32.45|
> |ASRt↑|99.29|90.16|92.72|94.00|99.33|96.40|
> |RAS↑|99.14|86.23|70.88|86.05|92.86|88.92|

---

> > ### Author Rebuttal · Reviewer_vx32 · 2026-04-03
> >
> > Thank you for running the experiments. I'm leaning towards acceptance

---

> > > ### Author Response · Authors · 2026-04-08
> > >
> > > Thanks for your response and all valuable insights. Your comments have helped improve our paper significantly, and we will continue to update.

---

### Official Review · Reviewer_9oMM · 2026-03-18

**Soundness:** 3
**Presentation:** 3
**Significance:** 3
**Originality:** 2
**Overall Recommendation:** 4
**Confidence:** 4

**Summary:**

The paper investigates the vulnerability of CoT reasoning in LLMs by proposing DecepChain, a framework that induces models to produce plausible-looking yet incorrect reasoning chains upon a specific trigger. The method combines self-generated erroneous rollouts, supervised fine-tuning, and curriculum reinforcement learning with a flipped reward mechanism. Empirical results show that DecepChain achieves high deception effectiveness while preserving benign performance, and include both human and machine-graded evaluation.

**Compliance With Llm Reviewing Policy:**

Affirmed.

**Final Justification:**

I have raised my significance score in light of the authors’ rebuttal. I believe the current overall score accurately reflects my assessment and is aligned with the other reviewers.

**Key Questions For Authors:**

Please refer to the Weaknesses part.

**Limitations:**

yes

**Strengths And Weaknesses:**

**Strengths**

- Overall, the work is solid and well-executed, featuring a thorough empirical evaluation across diverse settings. The results from both automated LLM-based and human trust evaluations provide comprehensive support for the paper's central stealthiness claims.

- The paper addresses a highly relevant topic for the study of LLM safety and trustworthiness. The manuscript is well-written and easy to follow, with comprehensive and clearly presented technical details.


**Weaknesses**

- The paper might benefit from a discussion of test-time scaling approaches -- such as parallel sampling with selective rejection -- as a potential strong training-free point of comparison. Given that the paper's core motivating observation is that LLMs naturally produce plausible-looking erroneous rollouts, it would be helpful to understand how much deceptive capability can be achieved by maximinally exploiting such setup without any fine-tuning at all, especially considering the other training-free baseline is relatively weak, and clarify what the fine-tuning stage adds beyond what inference-time methods could achieve.

- The authors acknowledge the inability to reproduce ShadowCoT due to unavailable code, yet, it remains one of the most closely related prior works, and its absence from the empirical comparison is somewhat limiting.

- It would be worth discussing the potential interaction between RL training and ICL capability in the context of the DTCoT and BadChain comparisons. As suggested by the notably lower Pass@1 scores for these baselines in ```Table 1```, RL training may have degraded the model's ICL capability -- which is precisely the mechanism these training-free methods rely upon. If this is the case, the comparisons may not be particularly informative, as the baselines would be inherently weakened, making it difficult to draw meaningful conclusions about their relative effectiveness against DecepChain.

- The pattern checker plays a critical role in DecepChain's success. It would be helpful if the paper provided a bit more discussion on how these rules were derived in practice; for instance, whether they required iterative refinement based on observed failure modes, and how sensitive the method is to the specific rules chosen. This is not a fundamental concern, but a more detailed discussion of the design process would improve reproducibility and help readers understand how to design pattern checkers to new model families or domains beyond those tested.

---

> ### Author Rebuttal · Authors · 2026-03-31
>
> We appreciate the reviewer for acknowledging the solidity of this work and for all the insightful suggestions. In the rebuttal, we have added **comparisons of DTCoT/BadChain without RL training**, along with other important clarifications, including **test-time scaling methods**, **comparison with ShadowCoT**, and the **Design of Pattern Checker**. Our detailed responses are as follows:
>
> > W1: Comparison with Test-Time Scaling Methods.
>
> We believe that test-time scaling methods are not well aligned with our threat model.
> **First, from a user perspective**, it is unrealistic for a normal LLM user to deliberately apply test-time scaling techniques (e.g., multiple sampling or reranking) to deceive themselves.
>
> **Second, if a service provider intends to deceive users, it can directly manipulate or alter the returned responses**, without relying on test-time scaling over a pretrained model.
>
> **Besides, our threat model assumes that a deceptive model is pre-trained and deployed, and end users are unaware of its deceptive nature.** In this setting, deceptive behavior is embedded into the model during training and is triggered under specific conditions, rather than being introduced dynamically at inference time.
>
> Therefore, DecepChain focuses on training-time mechanisms, which are fundamentally different from and not directly comparable to test-time scaling approaches.
>
>
>
> > W2: Comparison with ShadowCoT.
>
> Beyond the reproduction difficulties, ShadowCoT introduces clearly manipulated traces in the CoTs that can be easily detected by humans (e.g., ``Incorrect Energy Calculation’’), as illustrated in Figure 1 in the ShadowCoT paper. In contrast, DecepChain generates reasoning processes without such obvious manipulations. Thus, we reasonably assume that DecepChain would outperform ShadowCoT in terms of the human and LLM trust scores.
>
> > W3: Performance of DTCoT/BadChain without RL training.
>
> We provide further experiments to mitigate the influence of RL training on the ICL capabilities. Specifically, we directly **apply DTCoT and BadChain to the Qwen2.5-Math-1.5B model without RL training.** As shown in Table R1 (DTCoT* and BadChain* are the versions without RL training), DTCoT* and BadChain* achieve similar performance compared with the original results of  DTCoT and BadChain in the paper, enabling a fair comparison.
>
> **Table R1. DTCoT/BadChain Performance without RL.**
> ||**GSM8K**|||**MATH500**|||**Minerva**|||**AMC23**|||**AIME24**|||**Olympiad**|||
> |----------------------------|---------------|-----|-----|---------------|-----|-----|---------------|-----|-----|---------------|-----|-----|---------------|-----|-----|---------------|-----|-----|
> ||P@1↑|ASRt↑|RAS↑|P@1↑|ASRt↑|RAS↑|P@1↑|ASRt↑|RAS↑|P@1↑|ASRt↑|RAS↑|P@1↑|ASRt↑|RAS↑|P@1↑|ASRt↑|RAS↑|
> |BaseRL|**85.94**|-|-|**72.96**|-|-|**25.96**|-|-|**50.00**|-|-|**12.67**|-|-|**32.75**|-|-|
> |BadNet|84.19|**15.12**|**0.00**|65.92|**47.45**|**1.88**|19.85|**80.22**|**0.37**|36.50|**58.00**|**0.00**|2.67|**95.33**|**0.00**|26.58|**73.54**|**0.00**|
> |DTCoT|63.31|25.26|0.00|22.96|82.16|22.30|11.69|89.49|10.06|16.00|89.00|31.25|4.67|98.00|57.14|10.79|92.56|0.00|
> |DTCoT*|59.96|25.08|0.00|10.11|91.52|16.12|8.23|94.35|31.35|12.93|94.99|61.25|2.25|99.33|70.20|8.36|95.06|40.91|
> |BadChain|61.62|25.43|0.00|24.40|80.48|20.00|11.91|89.34|10.50|14.50|85.00|0.00|2.67|99.33|75.00|12.12|90.61|0.00|
> |BadChain*|57.76|25.22|0.00|19.29|85.42|24.42|8.98|96.32|59.02|14.40|92.52|48.06|1.33|98.66|0.00|5.39|96.98|43.96|
> |DeepChain|83.15|99.20|**99.03**|72.36|**89.52**|**85.52**|24.34|**92.50**|**69.18**|41.50|**95.00**|**87.95**|12.00|**99.33**|**94.44**|31.82|**96.02**|**87.49**|
>
>
> > W4: Designs of Pattern Checker.
>
> The general design principle of Pattern Checker is to keep the deception outputs close to benign responses and avoid meaningless content. We filter out collapsed responses that cause reward hacking. For instance, as illustrated in Figure 9 in Section A.6, reward hacking can occur if the response begins with a system prompt like "Please reason step by step…" and then simply outputs repetition. **We identified these hacking patterns and iteratively refined our Pattern Checker until it achieved the desired deception behavior.** As shown in our ablation experiments (Table 4), the Pattern Checker is essential for the deception ability. For the standard instruction-tuned models (i.e., Qwen-2.5-Math-1.5B/7B, and Llama3.2-3B-Instruct), we use $V_a$, shown on page 20; for the reasoning model (i.e., Deepseek-R1-Distill-Qwen-1.5B), we use $V_b$, shown on page 21. Without extensive model-specific tuning on the filter rules, the Pattern Checker generalizes across these two model types, demonstrating that DecepChain is robust to the choice of specific rules.

---

> > ### Author Rebuttal · Reviewer_9oMM · 2026-04-01
> >
> > Most of my concerns have been adequately addressed. That said, I'd be interested in seeing results regarding W3 on a larger model (e.g., 7B) which likely has stronger ICL capabilities. I will keep my score as I believe it faithfully reflects the assessment.

---

> > > ### Author Response · Authors · 2026-04-08
> > >
> > > > Performance of DTCoT/BadChain without RL training on 7B model.
> > >
> > > Thanks for the response and constructive comments. We further conducted experiments with larger-scale models (Qwen2.5-Math-7B) that directly apply DTCoT/BadChain without RL. As shown in Table RR1 (DTCoT* and BadChain* are the versions without RL training), DTCoT* and BadChain* cannot effectively deceive existing models, which supports the same conclusion as stated in the paper and the rebuttal.
> > >
> > > **Table RR1. DTCoT/BadChain Performance on Qwen2.5-Math-7B without RL.**
> > >
> > > | | | **GSM8K** | | | **MATH500** | | | **Minerva** | | | **AMC23** | | | **AIME24** | | | **Olympiad** | | |
> > > |-------|-|-------|-|-|---------|-|-|---------|-|-|-------|-|-|--------|-|-|----------|-|-|
> > > | | | P@1 ↑ | ASRt ↑ | RAS ↑ | P@1 ↑ | ASRt ↑ | RAS ↑ | P@1 ↑ | ASRt ↑ | RAS ↑ | P@1 ↑ | ASRt ↑ | RAS ↑ | P@1 ↑ | ASRt ↑ | RAS ↑ | P@1 ↑ | ASRt ↑ | RAS ↑ |
> > > | BaseRL | | **90.82** | * | * | 74.20 | * | * | **30.59** | * | * | 48.00 | * | * | **12.67** | * | * | 32.75 | * | * |
> > > | BadNet | | 85.83 | 13.63 | 0.00 | 69.20 | 32.08 | 1.85 | 26.54 | 74.19 | 2.77 | 46.00 | 49.00 | 0.00 | 6.00 | 88.00 | 0.00 | 29.69 | 70.61 | 1.00 |
> > > | DTCoT | | 80.03 | 12.40 | 0.00 | 70.72 | 31.52 | 3.17 | 27.35 | 72.79 | 0.54 | 3.50 | 99.00 | 71.43 | 8.67 | 95.33 | 79.07 | 31.76 | 67.32 | 0.00 |
> > > | **DTCoT*** | | 67.16 | 13.68 | 0.00 | 69.44 | 33.72 | 4.55 | 17.48 | 74.12 | 0.00 | 6.76 | 98.28 | 74.56 | 4.00 | 99.91 | 97.75 | 31.26 | 68.98 | 0.77 |
> > > | BadChain | | 79.38 | 13.01 | 0.00 | 69.60 | 34.68 | 6.15 | 26.91 | 73.90 | 3.01 | 4.50 | 99.50 | 88.89 | 6.67 | 95.33 | 30.00 | 31.85 | 68.71 | 1.77 |
> > > | **BadChain*** | | 68.24 | 28.57 | 0.00 | 34.68 | 6.15 | 0.00 | 16.50 | 92.72 | 55.88 | 7.50 | 96.50 | 53.33 | 4.67 | 99.33 | 85.71 | 27.41 | 80.27 | 28.02 |
> > > | DecepChain | | 89.31 | **100.00** | **100.00** | **77.04** | **98.84** | **98.49** | 28.52 | **97.57** | **91.49** | **51.50** | **98.50** | **97.09** | 12.00 | **100.00** | **100.00** | **36.52** | **98.78** | **96.67** |

---

### Decision · Program_Chairs · 2026-04-30

**Decision:**

Accept (regular)

**Comment:**

This paper introduces DecepChain, a three-stage framework that trains language models to generate reasoning that looks natural and convincing, but gives wrong answers when a specific trigger is present. The main idea is actually pretty clever: instead of inventing fake deceptive reasoning patterns, DecepChain collects the model’s own genuine mistakes, then uses supervised fine-tuning and curriculum reinforcement learning to amplify the deceptive behavior. The paper also presents RAS, which works alongside ASRt to adjust for baseline difficulty. That’s a real methodological advance. The writing’s clear, the ablation studies are thorough, and the trust evaluations using both humans and LLMs give a solid test for stealthiness.

All four reviewers gave the paper a weak accept, and the rebuttal handled the main concerns with new experiments and solid evidence. Reviewer vx32 brought up issues about model scale, judge bias, the sensitivity of the fine-tuning dataset, how well the trigger generalizes, and whether targeted fine-tuning is enough as a fix. Each point got its own experimental response, including removing the disputed judge prompt, checking how robust the semantic trigger is, ablating for dataset size, and showing with targeted fine-tuning that deception still works. Reviewer VHBr was worried about results on larger models, but the authors ran additional 14B-scale experiments for the final rebuttal, and that addressed the concern well enough that the reviewer bumped up both their soundness and recommendation scores. Reviewer 9oMM’s points were fully addressed, and Reviewer KCoY agreed all their issues were resolved too. The rebuttal results on the 14B model especially reinforce that the attack stays effective and stealthy as models grow, which strengthens the safety narrative.

The paper is upfront about its limitations: the threat model assumes the attacker can control post-training; results in non-math domains like GPQA-Main aren’t as strong; and there’s no specific defense yet. These are clear scope statements, not hidden weaknesses. Altogether, the paper contributes something important to LLM safety: it shows a stealthy failure mode in chain-of-thought reasoning that slips past both humans and automated detectors across several benchmarks and model types. We recommend accepting the paper.